# A Brief History of Cardiothoracic Surgical Critical Care Medicine in the United States

**DOI:** 10.3390/medicina58121856

**Published:** 2022-12-16

**Authors:** Rafal Kopanczyk, Nicolas Kumar, Amar M. Bhatt

**Affiliations:** 1Department of Anesthesiology, Division of Critical Care, The Ohio State University Wexner Medical Center, Columbus, OH 43210, USA; 2Department of Anesthesiology, Pain Medicine and Critical Care Medicine, Massachusetts General Hospital, Harvard Medical School, Boston, MA 02115, USA

**Keywords:** history of medicine, cardiothoracic surgery, intensive care medicine, cardiopulmonary bypass

## Abstract

Cardiothoracic surgical intensive care has developed in response to advances in cardiothoracic surgery. The invention of the cardiopulmonary bypass machine facilitated a motionless and bloodless surgical field and made operations of increasing complexity feasible. By the mid-1950s, the first successful procedures utilizing cardiopulmonary bypass took place. This was soon followed by the establishment of postoperative recovery units, the precursors to current cardiothoracic surgical intensive care units. These developments fostered the emergence of a new medical specialty: the discipline of critical care medicine. Together, surgeons and intensivists transformed the landscape of acute, in-hospital care. It is important to celebrate these achievements by remembering the individuals responsible for their conception. This article describes the early days of cardiothoracic surgery and cardiothoracic intensive care medicine.

## 1. Introduction

Cardiothoracic surgical intensive care has evolved in concert with advances in cardiac surgery. Although the first successful surgery on the heart is attributed to Dr. Daniel Hale Williams, who repaired a traumatic laceration to the right ventricle in 1893, cardiac surgery as we know it today began in the early 1950s with the development of the cardiopulmonary bypass machine [1,2]. By the end of the 1950s, successful cardiac surgeries under motionless and bloodless surgical fields were being performed in centers around the world. Along with them, new hospital units were established to address the constant, high-level care needed by these patients in the postoperative period. Coincidentally, a new specialty of critical care medicine (CCM) also emerged around the same time as a consequence of a polio epidemic, which laid the groundwork for the innovations needed to advance cardiac perioperative care. These developments yielded an explosion of new technologies, therapies, and surgeries, completely changing the landscape of acute care. The vision and ingenuity of the professionals involved in these monumental accomplishments gave rise to today’s cardiac surgery and cardiothoracic intensive care units (CT-ICUs). It is essential to celebrate these achievements by remembering and honoring the legacy of the individuals responsible for these innovations. This article describes the beginnings of modern cardiac surgery and critical care medicine, along with their effects on the development of cardiothoracic surgical critical care medicine (CT-CCM).

## 2. The First Days of the Cardiac Surgery Revolution

A motionless and bloodless surgical field was necessary to advance the discipline of cardiac surgery [1]. After a decade of some successful cardiac procedures, such as coarctation repair and patent ductus arteriosus (PDA) closure, the 1940s ended with a clear understanding that further progress would require a cardiopulmonary bypass machine that could facilitate an immobile and clear field [1]. The greatest obstacle to the development of such a machine was the lack of an artificial oxygenator [1]. Consequently, between 1950 and 1955, five centers worked on various iterations of an oxygenator [1]. Dr. John Gibbon developed a vertical screen oxygenator at Jefferson Medical College and added a DeBakey roller pump to complete the apparatus. With funding from International Business Machines (IBM), he used his heart–lung machine to successfully perform sham operations on dogs by 1952 [1]. Encouraged by the animal trials, he applied the technique in humans; unfortunately, three out of his first four patients died, with the sole survivor coming close to death due to oxygenator clotting [1]. Dispirited by the results, Gibbon decided to stop open-heart surgeries and reevaluate. Serendipitously, Gibbon was visited by another cardiac surgeon from the Mayo Clinic in 1952, with whom he shared blueprints of his machine. Dr. John Kirklin and a team of Mayo engineers modified the design and, starting in 1955, successfully operated on four out of eight patients [1,3]. Dr. Emerson Moffitt was the anesthesiologist working with the team, declaring the achievement “another Manhattan Project” [3,4]. By 1958, the Mayo-Gibbon heart–lung machine was commercially available from a manufacturer, Custom Engineering and Development [1]. Additionally, in 1955 and only 90 miles away from the Mayo Clinic, Dr. Lillehei of the University of Minnesota successfully operated on patients with a bubble oxygenator developed in his laboratory by Richard DeWall, one of the first open-heart perfusionists. Within a year, the Travenol Company commercialized the sterile, disposable DeWall bubble oxygenator [1]. These innovations launched the field of cardiac surgery as we know it today.

It was quickly recognized that significant perioperative complications were just as important to the patient’s well-being as the repair of cardiac pathology utilizing a heart–lung machine [1,3,5]. Morbidity observed in the early days of cardiac surgery would not surprise any anesthesiologist or surgeon, even today. For instance, postoperative coagulopathy and bleeding were frequently encountered [1]. Additionally, air embolisms were common, as well as equipment failures, and cannulation problems [1]. Furthermore, many surgeries ended tragically because of preoperative misdiagnoses. These included Gibbon’s first surgery on a human patient employing a heart–lung machine, where a 15-month-old patient diagnosed preoperatively with an atrial septal defect actually had a large PDA, a fact only recognized postmortem [1]. Surprisingly, today’s intraoperative transesophageal echocardiographic exams still uncover up to 27% of surgically relevant, undiagnosed pathologies [6,7,8]. Finally, given the extremely high risk of surgery, only patients with no other medical options would undergo these procedures, making the risk even higher. The high burden of morbidity required ongoing care beyond the operating theater and, in part, gave rise to the first cardiac surgical intensive care units [1,3,5].

The first intensive care unit dedicated to postoperative cardiac surgery patients opened on 2 October 1956 at Saint Mary’s Hospital, a Mayo Clinic affiliate, in Minnesota [3]. It was called the Postoperative Cardiovascular Unit and was led by Sister Elizabeth Gillis, a head nurse, until it was taken over by Dr. Dwight McGoon [3]. The unit consisted of 12 beds and 17 specially trained staff members—13 nurses and 4 aides [3]. Both pediatric and adult patients were admitted to this unit directly after surgery. Four Catholic nuns trained in nursing developed the initial protocols for postoperative care, which included three pathways: constant care; intermediate care; and regular care [3]. Constant care was reserved for patients requiring immediate postoperative attention. These patients had their vital signs, including “chest discharge”, checked every 15 min [3]. There were no digital probes or monitors at the time, so patients needed frequent, thorough visual assessments. Likewise, adequate oxygenation was ensured by inspection, and oxygen was delivered via mask or tent [3]. Given this tremendous workload and lack of modern motoring equipment, a nurse could only care for one patient at a time, repeating the required tasks and charting by hand. This regimented care relaxed once a patient was deemed appropriate for intermediate care, where patients were checked every hour, then every 2 h. Finally, with a physician’s permission, patients would be transferred to a nearby nursing floor for regular care [3].

Multidisciplinary involvement was key to the success of Mayo Clinic’s blossoming cardiac surgery program [3]. This success, in part, stemmed from the understanding of the unique needs of this patient population. The education of all patient-care team members, from physicians to dieticians, was viewed as essential. For example, in the mornings before the shift change, Dr. Kirklin would give lectures to the nurses about cardiac physiology, the specifics of the procedures, and cardiac care [3]. Anecdotally, Dr. McGoon, one of the first cardiac surgery intensivists, would spend up to 12 h at a time in the unit to monitor patients; reportedly, he would not even leave for meals [3]. Additionally, resident physicians arrived with each patient from surgery, ensuring the transition of care. This work ethic trickled down to other professionals, such as Rosemary White, a dietician responsible for maintaining a heart-healthy diet in this patient population [3]. Finally, the first seeds of palliative care in cardiac surgery were also sown, given the high mortality seen in the earliest days of cardiac surgery. A surgeon and a nurse would meet families in a private room to deliver tragic news if the patient died during surgery. One nurse recollected, “I remember one Sunday when there were three deaths. I had already met with two families and was on my way to visit with the third…” [3].

Cardiac surgery pioneers west of the Mississippi, Drs. John Osborn and Frank Gerbode, also described a need for specialized staff tending to postoperative open-heart surgery patients [5]. In 1956, practicing at Stanford University, they converted a large room into a daytime surgical recovery unit. They quickly recognized that postoperative patient outcomes depended significantly on the skills and experience of the nursing staff [5]. This realization resulted in an employment strategy aimed at hiring nurses experienced in postoperative recovery. Interestingly, this approach was confronted by hospital administrations as it required the establishment of full-time nurses paid by the hospital, as opposed to private-duty nurses, which was a popular practice at the time. Osborn’s sentiment describing the situation can probably be echoed by many physicians today: “It was a long and difficult political struggle” [5].

The early days of cardiac surgery intensive care did not leave much of a written record in the scientific literature, however [5]. Many advances in postoperative care of open-heart surgery patients were propagated by word of mouth. Although world-changing experiences and innovations were adopted ubiquitously, they infrequently made it to medical print. Instead, conversations were held at conferences, airports, or over the telephone, recollecting specific experiences and novelties [5]. Nevertheless, the importance of cardiac critical care was undeniably accepted at centers performing cardiac surgery, thus beginning the field of CT-CCM.

## 3. Emergence of Critical Care Medicine and Its Effect on Cardiothoracic Surgical Care

The development of postoperative cardiothoracic surgical care could not have happened without other medical advancements. Fortunately, the cardiac surgical revolution coincided with another monumental transformation that allowed for the further growth of acute, in-hospital care. The widespread introduction of positive-pressure ventilation (PPV) during a polio epidemic changed medicine as a whole and was the impetus for the establishment of a new specialty: critical care medicine.

Each year, 26 August marks Bjørn Ibsen Day, a medical hallmark quietly celebrated by far too few intensivists [9]. It commemorates the day that Dr. Ibsen, a Massachusetts General Hospital (MGH)-trained anesthesiologist practicing in Copenhagen in 1952, proposed using positive-pressure ventilation and tracheostomies to save patients with respiratory muscle paralysis from polio infections [10]. Although the concept of PPV was established and routinely utilized during surgery, the idea of using it on the wards was novel. During the polio epidemic in Denmark, it quickly proved to be lifesaving, decreasing mortality from respiratory failure from 87% to 31% [10]. Lessons learned from this period gave rise to the development of the new concept of “intensive care”, where severely ill individuals collectively resided in a designated area of a hospital, tended by nurses and physicians specialized in organ failure support; however, care should be taken not to forget the contributions of Florence Nightingale and Dr. Walter Dandy, each contributing to the general idea of a designated space for the seriously ill [10,11,12]. Soon thereafter, the ward use of mechanical positive-pressure ventilators began, starting with the third model of the Morch piston respirator in 1954, followed by Bennett and Bird’s intermittent positive-pressure valves and the Emerson ventilator [12,13]. By 1958, 25% of American hospitals had respiratory care units, totaling more than 300 beds nationwide, with the most influential opening in 1961 at MGH [14,15]. These innovations greatly aided the further extension of critical care and perioperative safety, including cardiothoracic surgical critical care.

The 1960s and 1970s marked the expansion of the technology and science of intensive care, including laboratory testing, invasive monitoring, mechanical ventilation, AC electrical defibrillation, transvenous pacing, cardiopulmonary resuscitation, and more [12]. A team of critical care anesthesiologists from MGH was especially instrumental in propagating these new practices. Based on their experiences in the respiratory intensive care unit (RICU), Drs. Pontoppidan, Hedley-Whyte, Laver, Bendixen, and Egbert published *Respiratory Care* in 1965, the first book of its kind, describing the clinical knowledge and research specific to the critically ill, quickly heralded by professionals in the field as the “red bible”, given its red cover [15]. This publication rapidly placed MGH at the forefront of intensive care in the 1960s. Renowned for its scientific expertise, MGH’s anesthesiology department started attracting other talented clinicians, including a National Institutes of Health (NIH)-funded anesthesiologist, Dr. Warren Zapol, who advanced the technology of extracorporeal membrane oxygenation (ECMO) in the 1960s and applied it to clinical practice in the 1970s [15]. These groundbreaking initiatives significantly affected care for all the critically ill, including the postoperative patient populations.

The rate at which new technologies and techniques were emerging was perhaps outpacing the abilities of many physicians to keep up and comfortably care for patients who required the newer methods [12]. Consequently, a new cadre of physicians including pulmonologists, anesthesiologists, and surgeons became the early intensivists [12]. This trend was initiated in 1963 at the Presbyterian University Hospital in Pittsburgh, where Dr. Peter Safar, an anesthesiologist and the father of cardiopulmonary resuscitation, opened the first subspeciality training program in intensive care in the newly formed department of anesthesiology [16]. These events culminated with the foundation of the Society of Critical Care Medicine (SCCM) in February 1971 and the subsequent publication of the first issue of its journal, *Critical Care Medicine*, in 1973 [17].

Although anesthesiologists comprised the majority of intensive care specialists in most of Europe and Australia, this was not the case in the United States, where a mix of disciplines converged to practice critical care medicine. This heterogeneity created training and credentialing questions and resulted in stakeholders from surgery, anesthesiology, pediatrics, and internal medicine attempting to form a unified Board of Critical Care Medicine in the early 1980s [14]. However, the initiative failed because of the lack of agreement on training qualifications. Consequently, in 1986, each specialty established its own critical care certificate, resulting in the fragmentation of intensive care [14]. These decisions, compounded by changing economic motivations, resulted in the specialty of internal medicine becoming the leading discipline training American intensivists, followed by surgery and, lastly, anesthesiology [14]. This resulted in a delayed adoption of intensivists by CT-CCM as inter-departmental struggles and the lack of a unified vision for CCM continued. The current CCM landscape is a direct reflection of the trends that were initiated in the 1980s.

The events surrounding the evolution of CCM had a great impact on the state of medicine, including cardiothoracic surgery and its postoperative care. Further innovations in cardiothoracic surgical techniques and improvements in patient outcomes would not have been possible without the advances in laboratory testing, diagnostic monitoring, mechanical ventilation, cardiopulmonary resuscitation, and many more. As with the innovations of Gibbon and Kirklin, it is important to treasure the ingenuity of the individuals responsible for the transformation of the care of critically ill patients.

## 4. Growing Pains of Cardiothoracic Surgical Critical Care Medicine

The management of cardiothoracic surgical intensive care units (CT-ICUs) remained mostly surgeon-driven for many decades. The model of care centered around attending surgeons rounding before or after scheduled operations with surgical trainees, consultants, trained nursing staff, and other technicians carrying out the dictated care plans [5,18]. In the 1970s, however, anesthesiologists began participating in CT-CCM in a growing number of academic centers around the country.

As a natural extension of intraoperative expertise and training, the trailblazing anesthesiologists entering CT-ICUs were often cardiovascular anesthesiologists by trade. The earliest participation of anesthesiologists occurred at MGH, with some anesthesiologists having postoperative involvement in patient care in the 1960s, surrounding the RICU innovations described previously [15]. However, this trend did not fully materialize until the second half of the 1970s. For example, Dr. Myer (Mike) Rosenthal began caring for the patients of a private Palo Alto Medical Clinic cardiothoracic surgical group within two years of his employment at Stanford University in 1975. He was trained in the Navy and, at that time, was one of the few physicians in the country able to interpret pulmonary artery catheter values [19,20]. Similarly, at Emory University, Dr. Donald Finlayson joined the faculty as a professor in 1976, where he eventually became the Director of Critical Care Medicine, including an 18-bed cardiothoracic ICU. Dr. Finlayson was succeeded in 1990 by a prominent cardiovascular anesthesiologist, Dr. James Ramsey [21]. Another cardiovascular and critical care anesthesiologist, Dr. Robert Sladen, practiced both crafts at Stanford University alongside Dr. Rosenthal between 1978 and 1986. He then joined Duke University, where he expanded the roles of Duke’s anesthesiologists to begin caring for postoperative cardiac patients at the Veterans Affairs (VA) hospital [20,22]. In 1997, Dr. Sladen was recruited by Columbia University, where he established a new division of critical care medicine and became the medical director of the cardiothoracic and surgical intensive care units [23]. At Columbia University, Dr. Sladen successfully integrated a multidisciplinary model into CT-CCM practice and developed the diplomatic rounding concept of *strategy and tactics*, in which surgeons and attending intensivists established the plan for the day, and the ICU team would carry it out. The gradual inclusion of anesthesiologists into CT-ICUs set the stage for what followed next: a paradigm shift from a surgeon-centric to a collaborative care model.

At the turn of the 21st century, the approach to postoperative cardiothoracic care started to change once again. First, evidence of improved outcomes emerged, verifying the importance of intensivists’ participation in CT-CCM [24,25,26,27,28]. Second, the complexities of care involving mechanical circulatory support and ECMO made a top-down style of management impractical [20]. Third, a new wave of cardiothoracic surgery leadership contributed to elevating the subspeciality. For example, Dr. Nevin Katz, a cardiothoracic surgeon with an appreciation for critical care, spearheaded the initiative of creating the Foundation for the Advancement of Cardiothoracic Surgical Care (FACTS-care), which included the Annual Perioperative and Critical Care Conference [18]. With its first meeting held in 2004, it has since been incorporated into The Society of Thoracic Surgeons, and continues to meet annually [29]. Similarly, the next generations of anesthesiologists continued their interests in CCM and CT-CCM, resulting in further engagement and growth. Now, up to 70% of current critical-care anesthesiologists practice CT-CCM [30]. Finally, the recent pandemics of 2009 and 2020 further expanded the interest of the medical community in therapies previously associated with CT-CCM. ECMO support in adults has expanded significantly over the last two decades, increasing the number of disciplines wanting to care for patients requiring this treatment [31], as illustrated by an internist Dr. Daniel Brodie, a current thought leader in the field of veno–venous ECMO [32].

The coronavirus pandemic, with its aftermath, is likely to open a new chapter in CT-CCM. As of 2022, the lesson on the importance of coalitions, collaborations, and unity could not be clearer. Given the stormy waters of the last few years, it is crucial to step back and re-evaluate the good and the bad and strive for further improvements and advancements. Learning about our history can be the first step in this process. Figure 1 illustrates a timeline of important events in the history of cardiothoracic surgical critical care medicine, and Table 1 lists other crucial events in the cardiothoracic surgery saga.

## 5. Conclusions

The early days of cardiac surgery and its postoperative intensive care depended on groundbreaking innovations and the ingenuity of individuals and their teams. The memory of those accomplishments should be cherished and often revisited. Moreover, the recognition of the historical distinction of this field is critical when evaluating goals for the future. As recognized by our predecessors, our field needs specialized training to address unique challenges specific to intensive care of open-heart surgery, multidisciplinary involvement, scientific research, and the diplomatic acumen needed to convince stakeholders of the importance of necessary changes. Our future largely depends on understanding the lessons of the past.

## Figures and Tables

**Figure 1 medicina-58-01856-f001:**
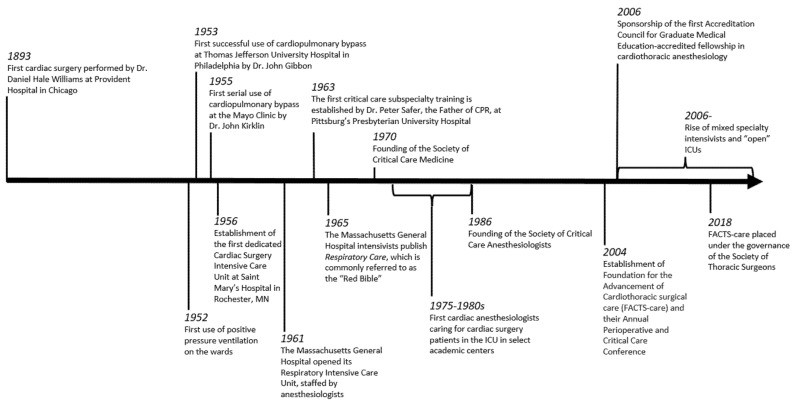
Timeline of important events for cardiothoracic surgical critical care medicine.

**Table 1 medicina-58-01856-t001:** List of achievements in cardiothoracic surgery and critical care.

Year	Innovator	Innovation	Affiliation	Impact
1893 [2]	Dr. Daniel Hale Williams	First cardiac surgery	Provident Hospital, Chicago, IL, USA	Sparked the realization that more sophisticated cardiac surgeries would require a method for immobilizing the heart.
1952 [10]	Dr. Bjørn Ibsen	Use of positive-pressure ventilation in ward patients with respiratory failure from polio	Kommunehospitalet, Copenhagen, Denmark	The first utilization of positive-pressure ventilation on the wards.
1953 [1]	Dr. John Gibbon	First successful use of cardiopulmonary bypass	Thomas Jefferson University Hospital, Philadelphia, PA, USA	Contributed to the creation of a motionless and bloodless surgical field.
1954–1955 [1]	Dr. C. Walton Lillehei	Controlled cross-circulation and bubble oxygenator	University of Minnesota	Developed the bubble oxygenator, which remained the standard for extracorporeal circulation until the 1970s.
1955 [1]	Dr. John Kirklin	First serial use of cardiopulmonary bypass	Mayo Clinic, Rochester, MN, USA	Contributed to the creation of a motionless and bloodless surgical field.
1956 [3]	Sister Elizabeth Gillis	Established the first Postoperative Cardiovascular Unit	Saint Mary’s Hospital, Rochester, MN, USA	Served as a model for subsequent dedicated cardiac surgery recovery units, the predecessors of modern CT-ICUs.
1963 [16]	Dr. Peter Safar	Established the first critical care subspecialty training	Presbyterian University Hospital, Pittsburg, PA, USA	Established the precedent for specialized critical care training for credentialing.
1966 [33]	Dr. Domingo Liotta and Dr. Michael DeBakey	First implantation of durable left ventricular assist device	Baylor College of Medicine, Houston, TX, USA	Expanded treatment options for patients with left ventricular failure as a bridge to heart transplantation or destination therapy.
1967 [34]	Dr. Christiaan Barnard	First human-to-human heart transplant	Groote Schuur Hospital, Cape Town, South Africa	Demonstration of the ultimate treatment for cardiomyopathy.
1970 [35]	Dr. Jeremy Swan and Dr. William Ganz	Development of the pulmonary artery catheter	Cedars-Sinai Medical Center, Los Angeles, CA, USA	Allowed for hemodynamic-driven therapy and enabled a rapid method for calculating cardiac output.
1977 [33]	Dr. Robert Bartlett	Published first report on serial use of ECMO	University of California, Irvine, Irvine, CA, USA	Demonstrated the efficacy of ECMO for various indications in both adult and pediatric populations.
1981 [34]	Dr. Norman Shumway and Dr. Bruce Reitz	First heart–lung transplantation	Stanford Medical Center, Stanford, CA, USA	Demonstration of the ultimate treatment for combined heart and lung disease.
1982 [33]	Dr. William DeVries, Robert Jarvik, and Dr. William Kolff	First artificial heart, the Jarvik-7, successfully implanted	University of Utah Medical Center	Expanded treatment options for patients with biventricular failure as a bridge to heart transplantation.
2002 [36]	Dr. Alain Cribier	First transcatheter aortic valve replacement performed	Hospital Charles Nicolle, Rouen, France	Greatly expanded treatment options for patients with severe aortic stenosis who were deemed poor candidates for open-heart surgery.
2022 [37]	Drs. Bartely P. Griffith and Muhammad M. Mohiuddin	First human heart xenotransplantation	University of Maryland, Baltimore, MD, USA	Potential expansion of organ pool for end-stage cardiomyopathy.

## Data Availability

Not applicable.

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
