# Peer review of "A Brief History of Cardiothoracic Surgical Critical Care Medicine in the United States"

_medicina, 2022, doi:10.3390/medicina58121856_

Round 1

Reviewer 1 Report

I would like to congratulate the authors on this interesting trip down history with the most important milestones which have lead to postoperative cardiac surgery care ase we know it. It is well written with interesting facts.

I would recommending adding "in the United States of America" to the title, since it is the main focus of this manuscript. Advances in other countries are not adressed. 

Author Response

We want to thank Reviewer # 1 for their comments. We have updated the title to indicate the location. Please see the tracked version with "in the United States" in red font. 

Reviewer 2 Report

The Manuscript by Kopanczyk et al demonstrated “A Brief History of Cardiothoracic Surgical Critical Care Medicine”. Authors writted the manuscript with the developmentals and ground-breaking innovations in the field of cardiothoracic surgery. Authors written the manuscript very well and should revise the manuscript with following comments before publication.

Major comments:

1.       A table with “inventor”, “innovation”, “Year” ,“affiliation”,“advantages”, “disadvantages”,  “reference” should give clear information about the developments in the CT surgery and critical care medicine.

2.       Authors should include the innovations made by Norman Edward Shumway who is a father of heart transplantation and had perfomed 800 heart tranplants.

3.       Authors should update the recent advances in CT surgery.

Author Response

We want to thank Reviewer #2 for their insightful comments and time reviewing the manuscript. 

 A table with “inventor”, “innovation”, “Year” ,“affiliation”,“advantages”, “disadvantages”,  “reference” should give clear information about the developments in the CT surgery and critical care medicine

  • Thank you for this comment. Please see below table with information desired. 

    Table 1. List of achievements in cardiothoracic surgery and critical care.

    Year

    Innovator

    Innovation

    Affiliation

    Impact

    1893[2]

    Dr. Daniel Hale Williams

    First cardiac surgery

    Provident Hospital, Chicago, IL, USA

    Sparked the realization that more sophisticated cardiac surgeries would require a method of immobilizing the heart.

    1952[10]

    Dr. Bjørn Ibsen

    Use of positive-pressure ventilation in ward patients with respiratory failure from polio

    Kommunehospitalet, Copenhagen, Denmark

    The first utilization of positive-pressure ventilation on the wards.

    1953[1]

    Dr. John Gibbon

    First successful use of cardiopulmonary bypass

    Thomas Jefferson University Hospital, Philadelphia, PA, USA

    Contributed to the creation of motionless and bloodless surgical field.

    1954-1955[1]

    Dr. C. Walton Lillehei

    Controlled cross-circulation and bubble oxygenator

    University of Minnesota

    Developed the bubble oxygenator which remained the standard for extracorporeal circulation until 1970’s

    1955[1]

    Dr. John Kirklin

    First serial use of cardiopulmonary bypass

    Mayo Clinic, Rochester, MN, USA

    Contributed to the creation of motionless and bloodless surgical field.

    1956[3]

    Sister Elizabeth Gillis

    Established the first Postoperative Cardiovascular Unit

    Saint Mary’s Hospital, Rochester, MN, USA

    Served a model for subsequent dedicated cardiac surgery recovery units, the predecessor of the modern CT-ICU.

    1963[16]

    Dr. Peter Safar

    Established the first critical care subspecialty training

    Presbyterian University Hospital, Pittsburg, PA, USA

    Established the precedent for specialized critical care training for credentialing.

    1966[33]

    Dr. Domingo Liotta and Dr. Michael DeBakey

    First implantation of durable left ventricular assist device

    Baylor College of Medicine, Houston, TX, USA

    Expanded treatment options for patients with left ventricular failure as bridge to heart transplantation or destination therapy.

    1967[34]

    Dr. Christiaan Barnard

    First human-to-human heart transplant

    Groote Schuur Hospital, Cape Town, South Africa

    Demonstration of the ultimate treatment for cardiomyopathy.

    1970[35]

    Dr. Jeremy Swan and Dr. William Ganz

    Development of the pulmonary artery catheter

    Cedars-Sinai Medical Center, Los Angeles, CA, USA

    Allowed for hemodynamic-driven therapy and enabled rapid method to calculated cardiac output.

    1977[33]

    Dr. Robert Bartlett

    Published first report on serial use of ECMO

    University of California, Irvine, Irvine, CA, USA

    Demonstrated the efficacy of ECMO for various indications in both adult and pediatric populations.

    1981[34]

    Dr. Norman Shumway and Dr. Bruce Reitz

    First heart-lung transplantation

    Stanford Medical Center, Stanford, CA, USA

    Demonstration of the ultimate treatment for combined heart and lung disease.

    1982[33]

    Dr. William DeVries, Robert Jarvik, and Dr. William Kolff

    First artificial heart, the Jarvik-7, successfully implanted

    University of Utah Medical Center

    Expanded treatment options for patients with biventricular failure as bridge to heart transplantation.

    2002[36]

    Dr. Alain Cribier

    First transcatheter aortic valve replacement performed

    Hospital Charles Nicolle, Rouen, France

    Greatly expanded treatment options for patients with severe aortic stenosis who were deemed poor candidates for open heart surgery.

    2022[37]

    Drs. Bartely P. Griffith and Muhammad M. Mohiuddin

    First human heart xenotransplantation

    University of Maryland, Baltimore, MD, USA

    Potential expansion of organ pool for end-stage cardiomyopathy

Authors should include the innovations made by Norman Edward Shumway who is a father of heart transplantation and had perfomed 800 heart tranplants.

  • Thank you for this comment. We included great contributions of Dr. Shumway in the table. 

Authors should update the recent advances in CT surgery.

  • Thank you for the comment. We included some recent advancements like xenotransplantation in the table. Given the topic of critical care history, it is out of the focus of this paper to include all the advances in cardiothoracic surgery.